# Potential Clinical Impact of LAFOV PET/CT: A Systematic Evaluation of Image Quality and Lesion Detection

**DOI:** 10.3390/diagnostics13213295

**Published:** 2023-10-24

**Authors:** Sabrina Honoré d’Este, Flemming Littrup Andersen, Julie Bjerglund Andersen, Annika Loft Jakobsen, Eunice Sanchez Saxtoft, Christina Schulze, Naja Liv Hansen, Kim Francis Andersen, Michala Holm Reichkendler, Liselotte Højgaard, Barbara Malene Fischer

**Affiliations:** 1Department of Clinical Physiology and Nuclear Medicine, Rigshospitalet, Copenhagen University Hospital, Blegdamsvej 9, 2100 Copenhagen, Denmark; 2Department of Clinical Medicine, Faculty of Health, Copenhagen University, Blegdamsvej 3b, 2200 Copenhagen, Denmark; 3School of Biomedical Engineering and Imaging Sciences, King’s College London, London SE1 7EH, UK

**Keywords:** whole-body PET, total-body PET, long-axial field-of-view (LAFOV), PET-CT, quadra, image quality, lesion detection

## Abstract

We performed a systematic evaluation of the diagnostic performance of LAFOV PET/CT with increasing acquisition time. The first 100 oncologic adult patients referred for 3 MBq/kg 2-[18F]fluoro-2-deoxy-D-glucose PET/CT on the Siemens Biograph Vision Quadra were included. A standard imaging protocol of 10 min was used and scans were reconstructed at 30 s, 60 s, 90 s, 180 s, 300 s, and 600 s. Paired comparisons of quantitative image noise, qualitative image quality, lesion detection, and lesion classification were performed. Image noise (n = 50, 34 women) was acceptable according to the current standard of care (coefficient-of-variance_ref_ < 0.15) after 90 s and improved significantly with increasing acquisition time (P_B_ < 0.001). The same was seen in observer rankings (P_B_ < 0.001). Lesion detection (n = 100, 74 women) improved significantly from 30 s to 90 s (P_B_ < 0.001), 90 s to 180 s (P_B_ = 0.001), and 90 s to 300 s (P_B_ = 0.002), while lesion classification improved from 90 s to 180 s (P_B_ < 0.001), 180 s to 300 s (P_B_ = 0.021), and 90 s to 300 s (P_B_ < 0.001). We observed improved image quality, lesion detection, and lesion classification with increasing acquisition time while maintaining a total scan time of less than 5 min, which demonstrates a potential clinical benefit. Based on these results we recommend a standard imaging acquisition protocol for LAFOV PET/CT of minimum 180 s to maximum 300 s after injection of 3 MBq/kg 2-[18F]fluoro-2-deoxy-D-glucose.

## 1. Introduction

The clinical application of PET/CT has been a fast-developing field, with long-axial field-of-view (LAFOV) PET being one of the most recent innovations. Since its development, PET has been implemented in clinical practice across a multitude of medical fields, and the advancements of various tracer compounds have enabled tracking of multiple biochemical processes in vivo [1,2,3,4,5,6]. The most widespread use of PET in current clinical practice remains within oncologic staging and treatment planning [7,8]. Inventions such as time-of-flight PET [9,10,11] and Lu-based scintillators with silicon photomultipliers have all contributed to the overall increase in sensitivity and clinical utility of PET/CT [12,13,14,15,16,17]. Although these technical advances have made progress in terms of image quality, conventional PET/CT remains limited by the increase of parallax error in the axial direction. Further, whole-body (WB) scans require multiple bed positions on a conventional PET/CT, increasing total scan time [18].

The 194-cm LAFOV scanner (uEXPLORER) was developed in 2018 by the University of California, Davis, California, USA (UC Davis) and United Imaging Healthcare (UIH), demonstrating faster imaging and improved image quality compared with earlier PET systems. The extended FOV secures an improved sensitivity, paving the way for lower injected activity and dynamic total-body PET [19]. Further, a WB scan on conventional PET/CT equipment would take approximately 12–20 min, while the uEXPLORER was able to achieve this in 1/10 of the time. In 2020, Siemens Healthineers launched the Biograph Vision Quadra PET/CT scanner with a LAFOV of 106 cm. The Biograph Vision Quadra and the conventional Biograph Vision 600 (FOV = 26.3 cm) were tested in a recent intra-individual comparison. The study found comparable measures of image quality, lesion quantification, and signal-to-noise-ratio on a 16 min conventional PET/CT device and in <2 min with LAFOV PET/CT, showing improved sensitivity and enabling ultra-fast and/or low-dose scanning [20]. Detailed physical characterization, according to the NEMA NU 2-2018 standard, of both the uEXPLORER [21] and the Biograph Vision Quadra [22] also showed increased sensitivity. Thus, adjustment of the current clinical imaging protocols should be implemented, as has already been stated in various studies [23,24,25,26,27,28]. With the potential for improved radiation protection, lesion detection, and general work-flow efficiency, clinical performance requires proper evaluation.

A Siemens Biograph Vision Quadra PET/CT scanner was installed at Rigshospitalet, Copenhagen University Hospital, Denmark, in September 2021. In this study, we evaluate the potential clinical benefit of LAFOV PET/CT and aim to inform a decision on the acquisition time in future clinical imaging protocols for LAFOV PET/CT. This is based on a systematic evaluation of image quality and lesion detection, i.e., number of image findings, with increasing acquisition time. Furthermore, certainty of diagnostic lesion classification, i.e., number of equivocally rated image findings, is evaluated.

## 2. Materials and Methods

### 2.1. Study Design

This is a retrospective evaluation of the image quality and lesion detection in oncology patients during the implementation phase (October 2021–December 2021) of a Siemens Biographs Vision Quadra in a tertiary hospital setting. The project was approved by the institutional review board on 17th September 2021, reference number 481_21. All patients provided written informed consent prior to inclusion.

### 2.2. Study Population

We included 50 consecutive patients for an evaluation of image quality. Additional patients were included, comprising a total of 100 patients for evaluation of lesion detection. The inclusion criteria were that all patients had to be over the age of 18, referred for clinical WB (skull-base to mid-thigh) 2-[^18^F]fluoro-2-deoxy-d-glucose (2-[^18^F]FDG) PET/CT on an indication of malignancy, and able to provide written, informed consent. Patients referred for treatment control of lymphoma were excluded from the evaluation of lesion detection due to the rarity of image findings in this patient group.

### 2.3. Imaging

An eGFR > 30 mL/min/1.73 m^2^ was confirmed before scanning and patients were required to have fasted for a minimum of 4 h prior to scanning, in accordance with clinical guidelines. A diagnostic PET/CT scan was performed with Visipaque 320 mg I/mL intravenous contrast (GE Healthcare AS, Chicago, IL, USA) after a 60 min rest post injection of 3 MBq/kg (2-[^18^F]FDG) (PET and Cyclotron Unit, Dept. Of Clinical Physiology and Nuclear Medicine, Rigshospitalet, Copenhagen, Denmark). PET data were acquired in compliance with EANM recommendations, using a standard clinical protocol [22] of 4 iterations with 5 subsets, maximum ring difference (MRD) of 85, 2 mm Gauss filter, 1.65 mm × 1.65 mm in plane voxel size, 2 mm slice thickness, point spread function, time-of-flight modeling, 106 cm fixed scan length achieved in one bed position (skull base to mid-thigh), and 600 s acquisition time with attenuation correction [29]. Data were acquired in a list-mode stream for 600 s and reconstructions were performed in reduced time intervals starting from time 0 to 30 s, 60 s, 90 s, 180 s, 300 s, and 600 s, respectively.

### 2.4. Evaluation of Image Quality

Evaluation of image quality was performed independently of evaluation of lesion detection. To enable complete blinding, the qualitative evaluation of image quality was carried out in Microsoft Powerpoint 2010. The quantitative evaluation was performed in syngo.via Client 8.3 (×64), MMoncology; Siemens Healthineers (syngo.via). For a quantitative evaluation, standard uptake value (SUV)_max_, SUV_mean_, and standard deviation (SD) were extracted from a spherical volume f interest (VOI), 60 mm in diameter, placed in the right liver lobe. The VOI was outlined on the 600 s reconstruction, aided by the CT data, avoiding disease, major vessels, or artefacts, and copied across all time points to ensure uniform placement. Coefficient of variance (COV) was computed as SD/SUV_mean_, with 0.15 considered equal to current standard of care of clinical PET/CT scans [29,30,31,32]. This is standard at Rigshospitalet. For a qualitative evaluation, two nuclear medicine physicians with 5 and 20 years of experience analyzing PET images respectively performed blinded rankings of patient reconstructions presented in random order. A random number generator was used to select certain scans to occur recurrently, to assess observer evaluation when presented with no variance in image quality. These scans replaced the 60 s and/or the 30 s reconstruction, since these were of least clinical interest in regards to visual differentiation. At no point were all image reconstructions of a particular subject available for side-by-side comparison. A ranging scale from one to six was used; one being the best image quality and six the lowest. Half a point was assigned if the observer evaluated the two images to be of the same quality. Each set of images included one maximum intensity projection (MIP) and three axial slices per reconstruction, presented as a slideshow, making DICOM header information unavailable. The axial slices were located at the following levels: carina of trachea, simultaneous display of liver, ventricle, and spleen, and lastly at the common iliac artery bifurcation.

### 2.5. Evaluation of Lesion Detection

All lesion detection was measured as number of image findings, and classification evaluation was performed using the reading application syngo.via. Sets of scans were randomly assigned to five nuclear medicine physicians all with more than 5 years of experience with PET/CT reading. All scans were blinded except for a shortened referral note, for instance: “Primary staging: Breast cancer”. A complete list of provided referral notes can be found in Appendix A. Each set of scans included five image reconstructions of 30 s, 90 s, 180 s, 300 s, and 600 s. They were viewed one at a time and always starting with the 30 s. Each set of scans was assessed by one observer (10–27 patient scans per observer). Each observer reported the total number of image findings on a pre-defined list of sites (Appendix A) for evaluation of lesion detection. The number of findings was noted, up to a maximum of five at each site. In addition, the image findings had to be characterized as “benign”, “equivocal”, or “malignant” for evaluation of lesion classification. “Benign” or “malignant” was applied when the observer was confident in the distinction. The classification “equivocal” was applied when observers were in doubt as to the pathogenesis of the lesion. The total count of image findings was divided into anatomical subgroups from the pre-determined list of sites: (1) organ, (2) lymph node, (3) ear–nose–throat, (4) thoracic, (5) abdominal, and (6) pelvic (Appendix A). The lesion detection of the reconstruction with image quality corresponding to a COV of 0.15 was applied as reference in order to compare with current clinical practice.

### 2.6. Statistical Analysis

All statistical analyses and data presentation were performed using Microsoft Excel 2013 and IBM SPSS Statistics 25. Statistical significance was achieved with a *p* value < 0.05. Bonferroni correction for all possible pairings was applied when multiple selective comparisons were performed. For the quantitative evaluation of image quality, paired comparisons of image noise computed as COV = SD/SUV_mean_ were carried out using a paired *t*-test with Bonferroni correction (P_B_) by a factor of 15. For the qualitative evaluation, inter-observer rankings were compared for each time point using Kappa statistics. Disagreement between observers was defined as any difference in the rating scales, half points included. The correlation between seconds of acquisition and image rank was examined using two-tailed Spearman’s Rho. A Wilcoxon signed-rank test was performed to examine potential differences between each time point for both observers (P_B_ by a factor of 30; 15 possible comparisons for each observer). For evaluation of lesion detection, Friedman’s test was performed to determine variation in image findings across acquisitions. Paired comparisons using Wilcoxon’s test were performed when Friedman’s test was significant (P_B_ by a factor of 70 for all possible comparisons, 10 for each subgroup, and 10 for the total number of image findings). Lesion classification was evaluated in the following time pairs: 90 s to 180 s, 180 s to 300 s, and 300 s to 600 s. McNemar’s test was applied to 2 × 2 contingency tables of image findings with binary outputs, noted as benign/malignant (BM) or equivocal (EQ), for the evaluation of lesion classification (P_B_ by a factor of 21).

## 3. Results

### 3.1. Demographics

For an evaluation of image quality, 34 women and 16 men, a total of 50 patients, with a mean age of 58 ± 17 years, were included. Post-injection time varied from 51–91 min. For an evaluation of lesion detection, 74 women and 26 men, a total of 100 patients, with a mean age of 59 ± 15 years, were included and post-injection time varied from 51–89 min. For complete demographics, please see Table 1.

There was an overlap of 37 patients between the image-quality population and the lesion-detection population after applying the exclusion criteria.

### 3.2. Exclusions

Inclusion and exclusion criteria of patients are depicted in Figure 1. A total of 107 patients were assessed for inclusion of 50 patients for the image-quality study (Figure 1a), and 205 for inclusion of 100 patients for the lesion-detection study (Figure 1b).

### 3.3. Image Quality

See Figure 2 for mean values of image noise as a function of acquisition time.

One patient out of 50 had a 30 mm rather than 60 mm VOI placed in the right liver lobe due to wide-spread disease and major vessels. Image noise computed as COV was closest to current acceptable values for standard of care after 90 s on the Vision Quadra (COV_mean_ = 0.14). Therefore, 90 s acquisition was implemented as the reference acquisition for current clinical standard in PET/CT in this study. As expected, COV improved significantly with increasing acquisition time across all time points (P_B_ < 0.001). Axial slices and MIP as viewed by observers in the qualitative evaluation can be seen in Figure 3.

One patient was excluded from qualitative analysis because the MIP rotation did not work in Microsoft Powerpoint 2010. In 14 out of 49 total sets of scans, the 60 s and/or the 30 s reconstructions were replaced with randomly selected repeats of other time points. This resulted in a total of n = 51 for 30 s, n = 35 for 60 s, n = 53 for 90 s, n = 50 for 180 s, n = 52 for 300 s, and n = 53 for 600 s images (n_total_ = 294). All scans were found to be of adequate visual image quality, except for one 30 s reconstruction by observer B. Observers achieved perfect agreement on the 30 s acquisition being ranked as of the worst quality (*k* = 1.00), substantial agreement for ranking of 60 s, 90 s, and 600 s (*k =* 0.73–0.85), and moderate agreement for 180 s and 300 s (*k =* 0.58–0.63) (*p* < 0.001). A strong negative monotonic correlation between acquisition time and observer rankings of ρ_A_ = −0.976 and ρ_B_ = −0.975 (observer A and B respectively) was observed (*p* < 0.001). A significant difference in rankings between each time point with increasing acquisition time was found for both observer A and observer B (P_B_ < 0.001). Distribution of change in rank, numerator, and denominator are depicted in Table 2.

### 3.4. Lesion Detection

See Figure 4 for examples of a single set of scans across acquisition times.

Significant variation in lesion detection was found across acquisition times for total findings (*p* < 0.001) and for all six subgroups (*p* ≤ 0.017). The total number of image findings significantly increased with increasing acquisition time from 30 s to 90 s, from 90 s to 180 s, and from 90 s to 300 s (P_B_ ≤ 0.002). No significant difference was found from 180 s to 300 s or from 300 s to 600 s. Likewise, no significant differences were found when dividing total image findings into those of lymph nodes and organ findings (P_B_
*=* 0.073–0.557). No significant difference was found between any of the performed acquisition comparisons in any of the four anatomical subgroups (P_B_ > 0.999). All *p* values for the performed comparisons in the evaluation of lesion detection can be found in Table 3.

The total number of equivocally rated image findings decreased significantly across all time pairings: 90 s to 180 s, 180 s to 300 s, and 90 s to 300 s (P_B_ ≤ 0.021). The number of equivocal organ lesions decreased significantly from 90 s to 180 s (P_B_ = 0.005) and from 90 s to 300 s (P_B_ < 0.001), while the number of equivocal lymph nodes only decreased significantly when comparing the 90 s to 300 s reconstructions (P_B_ = 0.003). The changes in the number of equivocal image findings were not statistically different when looking at the anatomical subgroups. All *p* values for the performed comparisons in the evaluation of lesion classification can be found in Table 4.

Figure 5 illustrates the total number of image findings as a function of acquisition time. Appendix A shows a complete list of numbers of image findings at individual sites and acquisition times.

## 4. Discussion

The purpose of this study was to perform a systematic evaluation of image quality and lesion detection dependent on acquisition time with long-axial field-of-view PET/CT. Among the many possibilities for optimizing imaging with a LAFOV PET/CT device, we chose to focus on acquisition time. While we acknowledge the many interesting aspects of e.g., delayed imaging and dynamic image acquisition for research purposes, we found the acquisition time to be a key parameter in a busy clinical setting. Furthermore, this reflects that in a patient population consisting of primarily adult cancer patients, image quality and high throughput are of more importance than reduction in injected activity. We found image noise to correspond to current acceptable values for standard of care after just 90 s and to improve further with increasing acquisition time, while not increasing total scan time beyond that of current clinical PET/CT scans. Additionally, we demonstrated a potential for clinical impact through an improvement in lesion detection [33,34,35,36] and a decrease in equivocal findings after 180 s to 300 s with LAFOV PET/CT.

Our results on image noise are in accordance with those of a previous study performed on the Vision Quadra, showing reduced noise with increasing acquisition time and producing PET scans of adequate image quality with acquisition time as low as 30 s [20]. That study (n_FDG_ = 20, n_18F-PSMA-1007_ = 16, n_68Ga-DOTA-TOC_ = 8) ranked the 600 s reconstruction as the highest quality in 100% of their blinded scans. In our results, rankings significantly improved with increasing acquisition time. However, in comparison to the previously mentioned study, only 86–93% (for observer A and B, respectively) of the 600 s reconstructions were ranked as the highest quality (*P_B_* < 0.001). This suggests that the observers were not always able to discriminate between 300 s and 600 s acquisition times. A similar pattern was observed when comparing the shorter acquisition times, where the number of equally rated scans lessened until the 180 s and 300 s acquisition times (A and B, respectively). The increase in equally rated scans beyond these time points suggests that visual distinction of image quality is harder past 180 s and 300 s, which aligns with our results for lesion detection.

The abovementioned study concluded that all target lesions could be identified on every LAFOV (30 s, 60 s, 120 s, 240 s, 600 s) PET/CT reconstruction as well as on the conventional PET/CT images analyzed [20]. In contradiction, our results showed that lesion detection improved from 30 s to 90 s to 180 s and from 90 s to 300 s, but no additional improvement in lesion detection is to be expected by increasing acquisition time beyond 300 s with injection of 3 MBq/kg 2-[^18^F]FDG. Although the lesion detection was not statistically different from 180 s to 300 s to 600 s, observers commented on the shorter time needed for reading as well as the reduced level of exhaustion with increasing acquisition time. While neither was systematically assessed, both translate into a higher throughput and lower risk of errors caused by burnout of physicians, which can help improve overall departmental efficacy. No difference in lesion detection over time was found when dividing the total image findings into those of organ lesions and lymph nodes, although notable improvements were observed in both, suggesting the study was underpowered. These results indicate that diagnostic performance for oncologic PET/CT can be improved with a short acquisition protocol of 3–5 min for LAFOV PET while still providing a faster total scan time than conventional PET/CT.

Our results on lesion classification showed improved diagnostic confidence, measured as a decrease in equivocally rated image findings, across all evaluated time pairings. This implies the potential clinical impact of the 300 s reconstruction, even though no significant difference in lesion detection was shown from 180 s to 300 s. These results suggest a preferred acquisition time, accommodating for high throughput, detection, and diagnostic confidence, in the range of 180 s to 300 s.

While no significant differences in lesion detection or classification were found within any of the anatomical subgroups, notable increases in image findings were seen in the abdominal/pelvic region from 90 s to 300 s and from 90 s to 180 s, possibly indicating relevance for further research and providing interest for future investigation into this diagnostically challenging region.

This study utilized an MRD of 85. Image noise would most likely have been reduced, had the full potential of all 322 rings been available at time of scan, which speaks for the strength of the study. Further strengths of the study are the consecutive inclusion of patients referred for PET/CT across a broad spectrum of malignant indications, the focus on 2-[^18^F]FDG, and the standardized imaging protocol, which is in line with the current EANM recommendations [29]. The patients presented in this study are representative of patients in clinical practice in many larger PET/CT centers. The broad inclusion criteria also served as a limitation; no significance could be proven within the anatomical sub regions due to the insufficient sample sizes of specific organ lesions. A larger study population could help shed light on how significant the difference from 180 s to 300 s might be, since our results imply the tipping point to be within this range. Although a study population of more than 100 patients would be beneficial, this also indicates high number-needed-to-treat (NNT), which reduces the likelihood of clinical relevance. Image findings were not corrected for false positives and were not individually numbered, complicating the process of lesion-by-lesion tracking for the paired comparison of lesion classification. This was accommodated for by the pre-defined list of 43 sites and the maximum of five findings per site. Quantitative image quality was not addressed at lesion level, in part due to the large number of lesions and dependence on lesion site. Reconstructions were generated through list-mode streaming from the start of the scan until the given acquisition time, not considering the biologic variation in physiological uptake or movement at a later time. We expect this potential error to be minimal during 10 min of scan time with patients strapped in following 60 min of rest, and thus, have not opted to perform an evaluation on impact of motion [37]. Although sampling from a subset of the full 600 s acquisition was possible, we believe the utilized method provides the better presentation. A few outliers resulted in a high variation in post-injection time; however, the mean ± SD remained at an acceptable interval, and thus, it did not appear to have had an impact on the results (Table 1). No control of histopathology was introduced for the lesion classification, limiting the results to clinical certainty of diagnostic classification.

The improvement in image quality provided by LAFOV PET has been tied to a potential reduction in injected tracer activity [19,20,38], while still achieving images of clinically acceptable quality [20,26]. The current applied activity from a single PET scan might not provide a high effective dose [39,40]; however, continuous follow-up PET scans and extended work with administration of radioactive tracers does result in increased radiation exposure over time. The implementation of low-dose scanning can help minimize this exposure, which will benefit both patients and technical personnel. The inevitable next step would be ultra-low-dose scans to make possible routine patient screenings and healthy human testing of tissue absorption in newly developed drugs. The improved image quality can also be translated into faster imaging protocols that increase patient comfort and can potentially reduce the need for sedation, i.e., in pediatric patients [41,42]. Likewise, shorter scan time will help to diminish psychological factors such as patient anxiety, stress, and claustrophobia, which remain some of the most uncontrollable and time-consuming factors in clinical practice. LAFOV PET/CT has also made a variety of advanced imaging feasible. This entails but is not limited to dual-tracer imaging, dynamic imaging with ongoing distribution in the body–organ interactions, as well as delayed imaging with improved differentiation of pathologic lesions as opposed to inflammation [43,44,45,46]. These might turn out to be the true benefit of LAFOV PET/CT. As future possibilities are being uncovered, one fact remains: LAFOV PET/CT continues to be far more expensive than conventional PET/CT scanners. Current and ongoing results have been promising; however, further research shedding light on the cost benefit and long-term impact on patient outcomes is needed. A total cost-containment analysis of all expenditures and all gains including benefit from new research results and from the use of pediatric patients is, however, a complex endeavor.

## 5. Conclusions

Long-axial field-of-view PET/CT enables adequate image quality after just 30 s and equals conventional clinically acceptable quality after 90 s. Lesion detection improved until 180 s acquisition time. Lesion classification further improved from 180 s to 300 s. Potential clinical impact was demonstrated through improved lesion detection and classification while keeping the total scan time within 5 min. Based on these results, we suggest a clinical imaging protocol for long-axial field-of-view PET/CT of minimum 180 s and maximum 300 s acquisition after injection of 3 MBq/kg 2-[^18^F]FDG.

## Figures and Tables

**Figure 1 diagnostics-13-03295-f001:**
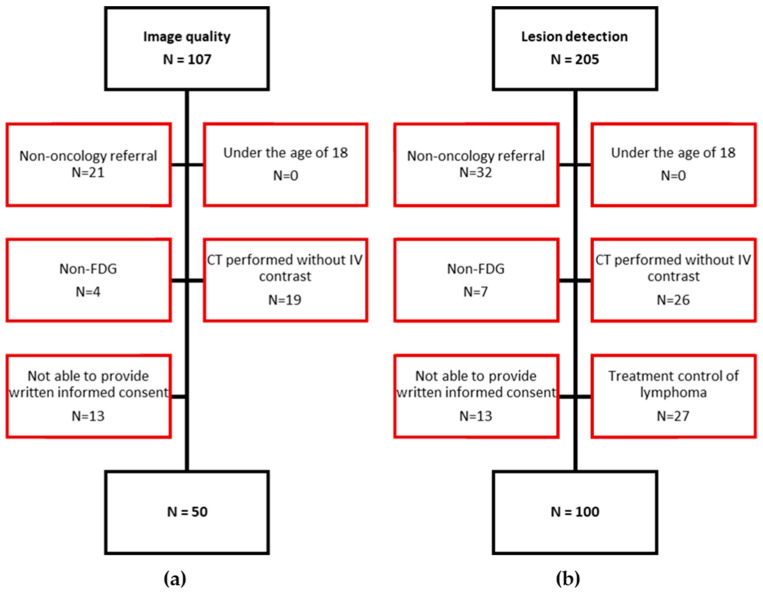
Flow diagram depicting in- and exclusion of study participants: (**a**) the evaluation of image quality and (**b**) the evaluation of lesion detection. (N) Number of patients; Red marks the excluded patients.

**Figure 2 diagnostics-13-03295-f002:**
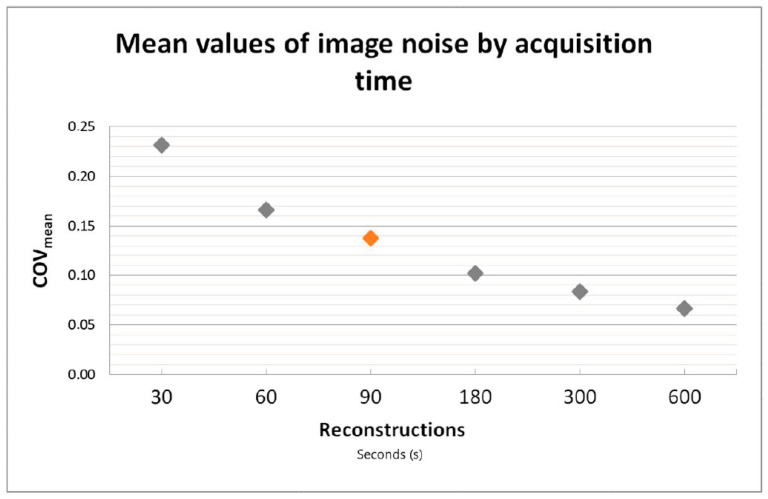
Coefficient of variance (COV) as a measurement for image noise presented as a function of acquisition time. COV was found to be closest to current PET/CT standard of care after 90 s on the Vision Quadra (marked in orange).

**Figure 3 diagnostics-13-03295-f003:**
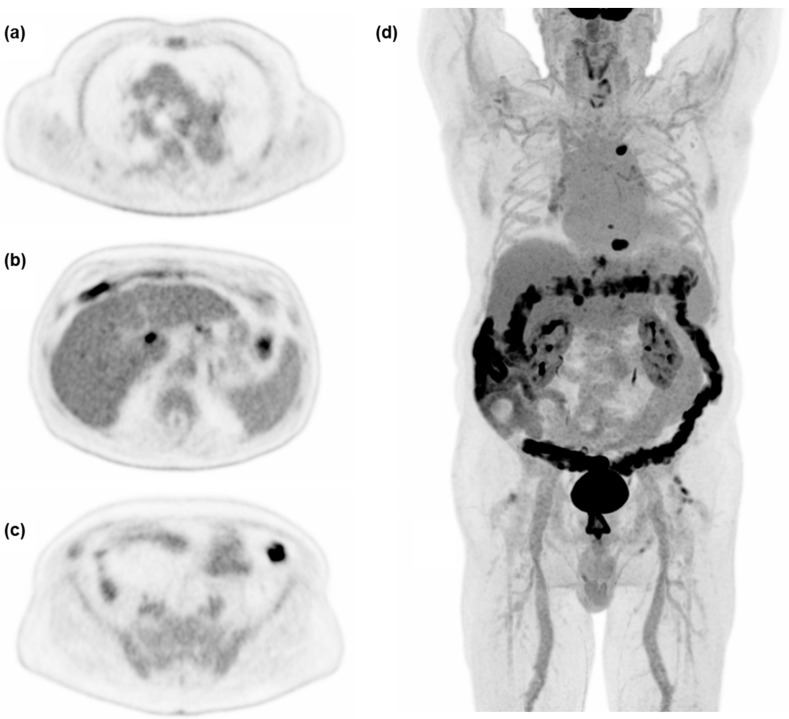
Examples of axial slices and maximum intensity projection (MIP) at 600 s acquisition time, used for the qualitative evaluation of image quality in a 75 y male referred for treatment control of hepatocellular carcinoma. (**a**) Level of the carina of the trachea, (**b**) level of simultaneous display of liver, ventricle, and spleen, (**c**) level of the common iliac artery bifurcation, (**d**) MIP.

**Figure 4 diagnostics-13-03295-f004:**
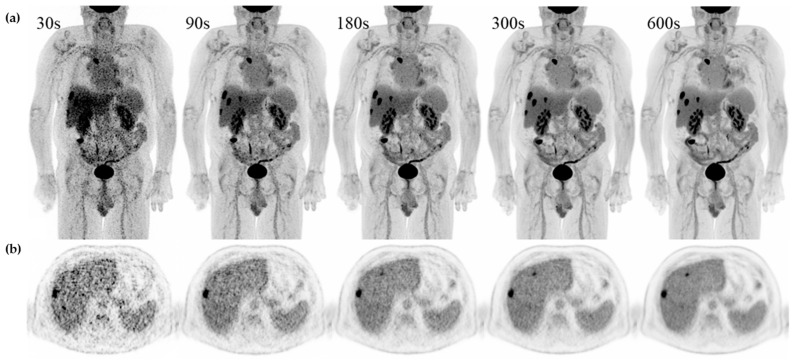
Examples of a single set of scans across acquisition times in a 71 y male referred for recurrent hepatic cancer. (**a**) Maximum intensity projection (MIP) and (**b**) axial slice with simultaneous display of liver, ventricle, and spleen from 30 s, 90 s, 180 s, 300 s, and 600 s from the same patient.

**Figure 5 diagnostics-13-03295-f005:**
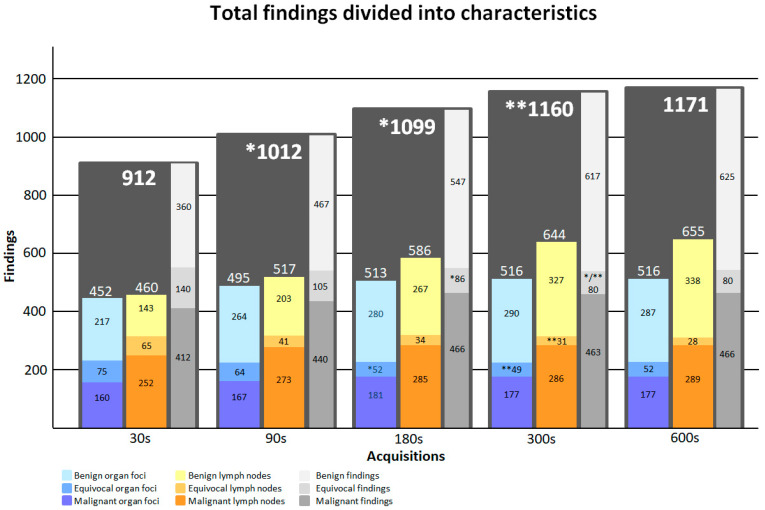
Image findings presented as organ lesions (blue scale), lymph nodes (yellow scale), and total numbers (grey scale). The distributions of benign, malignant, and equivocal findings are shown as a function of acquisition time for organ lesions, lymph nodes, and the total. *: Significant change from previous time-point value. **: Significant change from 90 s acquisition.

**Table 1 diagnostics-13-03295-t001:** Demographics.

Setting	Image Quality	Lesion Detection
No. of participants	50	100
Primary diagnosis	11	30
Staging	7	17
Treatment control	20	19
Evaluation of recurrence	12	34
Female/Male	34/16	74/26
Mean age (years) ± SD	58 ± 17	59 ± 15
Mean weight (kg) ± SD	76 ± 17	78 ± 19
Mean BMI (kg/m^2^) ± SD	26 ± 5	27 ± 7
Post-injection wait time (min) ± SD	65 ± 8	67 ± 8

Note: Demographics table showing distribution of characteristics for the evaluation of image quality and the evaluation of lesion detection and classification. BMI: body mass index; SD: standard deviation.

**Table 2 diagnostics-13-03295-t002:** Distribution of change in rank presented in percentages for every paired comparison for both observers.

Setting & Comparison	30 s → 60 s	60 s → 90 s	90 s → 180 s	180 s → 300 s	300 s → 600 s
Observer A					
Increase in rank (n)	94% (33/35)	94% (33/35)	96% (52/54)	89% (47/53)	86% (48/56)
Equal rank (n)	6% (2/35)	6% (2/35)	2% (1/54)	11% (6/53)	11% (6/56)
Decrease in rank (n)	0% (0)	0% (0)	2% (1/54)	0% (0)	4% (2/56)
P_B_	<0.001	<0.001	<0.001	<0.001	<0.001
Observer B					
Increase in rank (n)	94% (33/35)	97% (34/35)	94% (51/54)	98% (52/53)	93% (52/56)
Equal rank (n)	6% (2/35)	3% (1/35)	2% (1/54)	0% (0)	7% (4/56)
Decrease in rank (n)	0% (0)	0% (0)	4% (2/54)	2% (1/53)	0% (0)
P_B_	<0.001	<0.001	<0.001	<0.001	<0.001

Significance (P_B_) is Bonferroni corrected by a factor 15. Due to the repeated requisitions, the total number of comparisons exceeds the total number of images. Significant difference in rankings between chosen time points was found for both observers. The percentage of equal rated requisitions decreased until 180 s for observer A and until 300 s for observer B. n: Number of patient scans in the given category.

**Table 3 diagnostics-13-03295-t003:** Difference in number of lesions and patients with lesions (total and regional) diagnosed on the respective reconstruction.

Setting and Comparison	30 s → 90 s	90 s → 180 s	180 s → 300 s	300 s → 600 s	90 s → 300 s
Total image findings					
n_Total 1_ → n_Total 2_	95 → 95	95 → 96	96 → 98	98 → 98	95 → 98
Difference in findings	+100 *	+87 *	+61	+11	+148 *
P_Wilcoxon_	<0.001	<0.001	0.002	0.166	<0.001
P_B_	<0.001	0.001	0.150	-	0.002
Lymph nodes					
n_LN 1_ → n_LN 2_	62 → 66	66 → 70	70 → 72	72 → 72	66 → 72
Difference in findings	+57	+69	+58	+11	+127
P_Wilcoxon_	0.002	0.002	0.003	0.134	0.002
P_B_	0.148	0.152	0.230	-	0.154
Organ lesions					
n_Organ 1_ → n_Organ 2_	89 → 93	93 → 95	95 → 97	97 → 96	93 → 97
Difference in findings	+43	+18	+3	0	+21
P_Wilcoxon_	0.001	0.003	0.565	>0.999	0.008
P_B_	0.073	0.229	-	-	0.557
Ear–nose–throat					
n_ENT 1_ → n_ENT 2_	32 → 33	33 → 36	36 → 36	36 → 36	33 → 36
Difference in findings	+10	+20	+9	+2	+29
P_Wilcoxon_	0.066	0.066	0.285	0.157	0.109
P_B_	-	-	-	-	-
Thorax					
n_Thorax 1_ → n_Thorax 2_	30 → 35	35 → 35	35 → 35	35 → 36	35 → 36
Difference in findings	+16	+6	+1	+3	+7
P_Wilcoxon_	0.039	0.180	0.655	0.317	0.593
P_B_	>0.999	-	-	-	-
Abdomen					
n_Abdomen 1_ → n_Abdomen 2_	35 → 42	42 → 42	42 → 44	44 → 43	42 → 44
Difference in findings	+11	+11	+3	0	+14
P_Wilcoxon_	0.068	0.071	0.396	>0.999	0.041
P_B_	-	-	-	-	>0.999
Pelvis					
n_Pelvis 1_ → n_Pelvis 2_	40 → 44	44 → 46	46 → 47	47 → 46	44 → 47
Difference in findings	+7	+10	+5	0	+15
P_Wilcoxon_	0.102	0.026	0.102	>0.999	0.026
P_B_	-	>0.999	-	-	>0.999

* The image findings are compared according to location with corresponding *p* values. Bonferroni-corrected *p* values by a factor of 70, provided when primary *p* value was <0.05. Significance marked with *. n: Number of patients with lesions (total and regional) out of 100 possible diagnosed on the respective reconstruction, P_B_: Bonferroni-corrected *p* value, ENT: ear–nose–throat.

**Table 4 diagnostics-13-03295-t004:** Change in classification from equivocal lesion to benign or malignant lesion across compared acquisition times.

Setting & Comparison	90 s → 180 s	180 s → 300 s	90 s → 300 s
Total image findings			
Findings compared	1005	1088	998
EQ → BM	27 *	11 *	33 *
P_McNemar_	<0.001	0.001	<0.001
P_B_	<0.001	0.021	<0.001
Lymph nodes			
Findings compared	517	585	517
EQ → BM	14	5	17 *
P_McNemar_	0.004	0.063	<0.001
P_B_	0.088	-	0.003
Organ lesions			
Findings compared	488	503	481
EQ → BM	13 *	6	16 *
P_McNemar_	<0.001	0.031	<0.001
P_B_	0.005	0.656	<0.001
Ear-Nose-Throat			
Findings compared	117	136	117
EQ → BM	4	0	4
P_McNemar_	0.125	>0.999	0.125
P_B_	-	-	-
Thorax			
Findings compared	114	118	112
EQ → BM	3	0	3
P_McNemar_	0.250	>0.999	0.250
P_B_	-	-	-
Abdomen			
Findings compared	160	171	159
EQ → BM	4	3	3
P_McNemar_	0.125	0.250	0.250
P_B_	-	-	-
Pelvis			
Findings compared	117	126	116
EQ → BM	4	3	6
P_McNemar_	0.250	0.500	0.063
P_B_	-	-	>0.999

* The image findings are compared according to location, with corresponding *p* values. Bonferroni-corrected *p* values by a factor of 21, provided when primary *p* value was < 0.05. Significance marked with *. EQ: equivocal findings, BM: benign/malignant findings, P_B_: Bonferroni-corrected *p* value.

## Data Availability

Data generated or analyzed during the study are available from the corresponding author upon reasonable request.

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
