# Peer review of "Potential Clinical Impact of LAFOV PET/CT: A Systematic Evaluation of Image Quality and Lesion Detection"

_diagnostics, 2023, doi:10.3390/diagnostics13213295_

Round 1

Reviewer 1 Report

This manuscript attempted to highlight the optimal PET scanning time using the new LAFOV-PET/CT technology. This technology, based on various studies, has already demonstrated advantages both in terms of acquisition timing (reduction of times) and the dose administered to the patient (dose reduction). By studying oncology patients undergoing PET with FDG, this study highlights that the optimal acquisition time could be between 3 and 5 minutes. This time would certainly bring an advantage to common clinical practice even if at the moment, given the high prices of this technology, it is not clear what the cost-benefit ratio is compared to the technology currently in use.

I believe that this study is well constructed and highlights an aspect that could have a significant impact on clinical practice.

the level of English is appropriate.

Minor comments:

At line 15 I suppose that “was “ need to be changed in “were”

At line 59 I suggest to insert this recent study as citation: https://doi.org/10.1007/s40336-023-00547-7

AT line 102 SD must be explained entirely,

At line 315 I suggest to insert “differences” after “no significance”

At line 331 separate “than100” in “than 100”

Author Response

Dear reviewer,

We thank you for your time in reviewing our manuscript and the kind words. Please find our reply to your review report in the following.

  • At line 15 I suppose that “was “ need to be changed in “were”
    1. This has been corrected.
  • At line 59 I suggest to insert this recent study as citation: https://doi.org/10.1007/s40336-023-00547-7
    1. We thank you for this reference and have added the study as citation.
  • At line 102 SD must be explained entirely,
    1. This is an oversight on our part and has been corrected.
  • At line 315 I suggest to insert “differences” after “no significance”
    1. This has been corrected.
  • At line 331 separate “than100” in “than 100”
    1. This has been corrected.

Sincerely,

Sabrina Honoré d’Este

Reviewer 2 Report

This is an interesting article performing a systematic evaluation of image quality and lesion detection dependent on acquisition time for optimizing imaging with a LAFOV PET/CT. Some concerns are following:

1.When refered to evaluation of PET image quality, only one index discussed in this article,COV, is not enough.Moreover,as a standard protocol,phantom study is needed and recovery coefficients should be calculated. 

2.In abstract (Line 13-14), the author declared that "We aim to perform a systematic evaluation of the diagnostic performance of LAFOV 13 PET/CT with increasing acquisition time" ,but in maintext (Line 67), the aim of study turn to "determine image quality with decreasing acquisition time", please unified the statement

3.Line 104-105, connection symbols in "Volume-Of_Interest" is incorrect.

4.The description of P value is not unified for the whole manuscript, for example, in Line 231,there are two P value,P<.001 and P≤0.017. 

 Minor editing of English language required

Author Response

Dear reviewer,

We thank you for your time in reviewing our manuscript. Your concerns are highly appreciated and we have tried to accommodate them to the best of our ability. Please find our reply to your review report in the following.

  • When refered to evaluation of PET image quality, only one index discussed in this article,COV, is not enough.Moreover,as a standard protocol,phantom study is needed and recovery coefficients should be calculated.
    1. Thank you for your comment. COV is the standard common measure of image noise as it allows for easy comparison between subjects and reconstructions by sampling a homogenous region like the liver. We agree that other metrics might support the manuscript, for example contrast to noise ratio of lesions ( Mean(Lesion)-Mean(Background) / SD(Background) ), but these measures become very depending on lesion site and therefore hard to compare but also due to the large number of lesions in this study. We have added a sentence to our limitations to address this point.

“Quantitative image quality was not addressed at lesion level, in part due to the large number of lesions and dependence of lesion site”.

  1. In regards to the need for phantom study and recovery coefficients we refer to the data of the following citation table 6. As mentioned in the methods section a standard protocol of 4i5s psf+tof is used. We have included the study as citation in the Imaging paragraph of the methods section, where a standard protocol of 4i5d PSF+TOF is mentioned.

https://jnm.snmjournals.org/content/63/3/476/tab-figures-data

Further we have changed the phrasing of the conclusion to:

“…we suggest a clinical imaging protocol for…

  • In abstract (Line 13-14), the author declared that "We aim to perform a systematic evaluation of the diagnostic performance of LAFOV 13 PET/CT with increasing acquisition time" ,but in maintext (Line 67), the aim of study turn to "determine image quality with decreasing acquisition time", please unified the statement.
    1. We have made the following changes to the paragraph to unify the statements:

“This will be based on a systematic evaluation of image quality and lesion detection, i.e. number of image findings, with increasing acquisition time. Further, certainty of diagnostic lesion classification, i.e. number of equivocally rated image findings, will be evaluated.”

  • Line 104-105, connection symbols in "Volume-Of_Interest" is incorrect.
    1. This has been corrected.
  • The description of P value is not unified for the whole manuscript, for example, in Line 231,there are two P value,P<.001 and P≤0.017.
    1. This has been corrected throughout the manuscript.

Sincerely,

Sabrina Honoré d’Este